# The Effect of Process Conditions on Sulfuric Acid Leaching of Manganese Sludge

**DOI:** 10.3390/ma16134591

**Published:** 2023-06-25

**Authors:** Jafar Safarian, Ariel Skaug Eini, Markus Antonius Elinsønn Pedersen, Shokouh Haghdani

**Affiliations:** Department of Materials Science and Engineering, Norwegian University of Science and Technology, 7034 Trondheim, Norway; arielse@stud.ntnu.no (A.S.E.); mapeders@stud.ntnu.no (M.A.E.P.); shokouhhaghdani@gmail.com (S.H.)

**Keywords:** manganese, sludge, zinc, lead, potassium, sulfuric acid, ferromanganese, submerged arc furnace, kinetics, manganese carbonate

## Abstract

Manganese sludge, an industrial waste product in the ferroalloy industry, contains various components and holds significant importance for sustainable development through its valorization. This study focuses on characterizing a manganese sludge and investigating its behavior during sulfuric acid leaching. The influence of process conditions, including temperature, acid concentration, liquid to solid ratio, and leaching duration, was examined. The results revealed that Mn, Zn, and K are the main leachable components, and their overall leaching rates increase with increasing temperature, liquid to solid ratio, and time. However, the acid concentration requires optimization. High leaching rates of 90% for Mn, 90% for Zn, and 100% for K were achieved. Moreover, it was found that Pb in the sludge is converted to sulfate during the leaching, which yields a sulfate concentrate rich in PbSO_4_. The leaching process for Mn and Zn species appears to follow a second or third order reaction, and the calculation of rate constants indicated that Mn leaching kinetics are two to five times higher than those for Zn. Thermodynamic calculations were employed to evaluate the main chemical reactions occurring during leaching.

## 1. Introduction

Manganese (Mn) is an important element widely employed in both ferrous and non-ferrous metallurgy, mainly in its metallic form, to produce various alloys. The compounds of Mn have numerous applications, and one particularly significant use is the incorporation of manganese IV oxide (MnO_2_) in battery manufacturing. The main processing route for manganese ores involves their utilization in the production of manganese ferroalloys through a carbothermic reduction process in a submerged arc furnace (SAF) [1]. Previous studies have investigated the thermochemistry of the SAF process and the underlying reaction mechanisms involved in the production of Mn metal [2]. In the SAF process, the furnace charge mainly consists of Mn oxides, which are largely transformed into the Mn ferroalloy product after reduction. In addition, a smaller portion of Mn ends up being processed into slag as a byproduct [3]. Furthermore, the off-gas furnace carries fine particles (dust), which are separated from the gas using dedusting techniques [4,5]. When wet scrubbers are employed to capture the dust, the wet particles are separated from the gas through water spray, resulting in the formation of a sludge known as Mn sludge [5]. This Mn sludge contains significant amounts of valuable elements, including Mn, Zn, and Pb, making its valorization crucial for sustainable development and the circular economy. A study by Shen et al. [6] has characterized manganese furnace dust regarding the charge materials and indicated significant amounts of Mn and Zn in the dust. It is worth noting that the recycling of Mn sludge into the SAF process is difficult due to the presence of volatile components, such as Zn and K. Currently, this sludge is considered an industrial hazardous waste and is landfilled, emphasizing the importance of its valorization. Pyrometallurgical processing of SAF dust/sludge to recycle it into the SAF process has technical and economic limitations. However, there is growing interest in the application of hydrometallurgical processes, which offer potential profitability through the extraction of valuable metals [7].

Extensive research has been conducted on the utilization of hydrometallurgical processes for extracting Mn from both primary and secondary raw materials. The production of electrolytic Mn metal (EMM) and electrolytic Mn dioxide (EMD) through hydrometallurgical processing of Mn ores are mature processes. As mentioned earlier, the Mn yield in the SAF process is relatively low, with a portion of Mn lost mostly in the slag byproduct and dust. Therefore, there has been considerable interest in recovering Mn from these streams. Various studies in the literature have investigated the recovery of Mn from Mn-containing slags using different acids, such as H_2_SO_4_ [8,9,10,11,12,13]. Notably, high Mn recovery rates of 88% and 90% have been reported. Although the generation of Mn sludge is smaller compared to the slag byproduct in the SAF process, it represents a more complex material. The Mn sludge contains a significant amount of Mn, along with various highly volatile substances found in the SAF charge, such as Zn, Pb, K, Cd, etc. Sancho et al. [14] studied the extraction of electrolytic Mn metal from Mn sludge. Their process involved primary leaching using H_2_SO_4_, purification steps, and an electrolysis phase for electrolytic manganese production, resulting in the production of high-purity Mn metal. Similarly, Ghafarizadeh et al. [15] focused on the reductive leaching of SAF dust from the ferromanganese process using sulfuric acid, with additional reducing agents, such as oxalic acid, hydrogen peroxide, and glucose. They observed a Mn leaching rate of up to 40% using 0.5 to 3 M H_2_SO_4_ solutions at 70 °C within 90 min, with a liquid to solid ratio (L/S) of 10. Furthermore, they found that the presence of oxalic acid and hydrogen peroxide significantly increased the Mn leaching recovery (almost complete recovery), while glucose had a lesser effect. Sadeghi et al. [16] conducted the reductive acid leaching of a Mn residue using 1 M H_2_SO_4_. They employed glucose as the reducing agent, which resulted in high leaching rates of over 90% for Mn and Zn. Additionally, they observed that microwave-assisted leaching indicates a faster leaching rate compared to ultrasound-assisted and conventional leaching techniques.

In the present study, the leaching behavior of a Mn sludge using H_2_SO_4_ solutions is investigated through experimental work, aiming to evaluate the effect of process conditions. In hydrometallurgical processes used to produce Mn and Zn metals, sulfuric acid has been used as the leaching agent. Since the Mn sludge has a significant amount of these two metals, in the present study, sulfuric acid leaching was considered as the leaching agent to use. Various materials characterization techniques are employed to gain insights into the Mn sludge. Furthermore, thermodynamic calculations are performed to further understand the acid leaching process of Mn sludge.

## 2. Materials and Methods

### 2.1. Materials and Preparation

The manganese sludge used in this study was received from the ferromanganese industry and contained a significant amount of water. To minimize errors in mass measurement, the sludge was dried in an oven at approximately 90 °C for a period of two days. Sulfuric acid (H_2_SO_4_) solutions with concentrations of 0.7, 1.2, and 1.7 molar (M) were prepared using the standard methods of acid dilution by adding distilled water. Additionally, concentrated sulfuric acid (98.3% H_2_SO_4_) was diluted.

### 2.2. Leaching Experiments

The leaching experiments were performed in a beaker using a magnetic stirrer with a rotation speed of 350 rpm on a hot plate. Once the H_2_SO_4_ solution reached the target temperature, the Mn sludge powder was gradually added to the stirred solution. In all experiments, a precise amount of 10 ± 0.05 g of dried sludge powder was used. Different volumes of leaching solutions, namely 12.5, 25, 50, and 100 mL, were employed to achieve liquid to solid ratios (L/S) of 1.25, 2.5, 5, and 10 mL/g, respectively. To investigate the effect of temperature, six experiments were conducted at 30 °C and 50 °C, employing three different L/S ratios under fixed leaching conditions using a 1.7 M H_2_SO_4_ solution. The remaining experiments were carried out at 50 °C. To study the impact of leaching duration, leaching tests were performed for durations ranging from 10 to 120 min using 1.2 and 1.7 M H_2_SO_4_ solutions. After the leaching process, the leaching residue was separated by filtration using a Büchner funnel. The residue was then washed with distilled water, dried in an oven, and its mass was measured. The pH of the solution was measured for selected experiments both before and after the leaching process.

### 2.3. Characterization of Materials and Products

The chemical compositions of the dried sludge powder and leaching residues were measured using X-ray fluorescence (XRF) with a device from Thermo Fisher, Degerfors, Sweden. For XRF testing, sample preparation was conducted using the flux fusion method. Phase analysis of these materials was performed by X-ray diffraction (XRD) using the Bruker D8 A25 DaVinciTM, Karlsruhe, Germany, with CuK radiation (wavelength of 1.54 Å). The measurement range was set from 10 to 80° with a step size of 0.03. It is worth mentioning that XRD analysis was employed to study all the leaching residues. In addition, selected leaching solutions were analyzed using inductively coupled plasma-mass spectrometry (ICP-MS). Furthermore, the leaching residue samples were characterized by scanning electron microscope (SEM) (Zeiss Ultra FESEM, National Institute of Standards and Technology, Gaithersburg, MD, USA) equipped with an XFlash^®^ 4010 Detector supplied by the Bruker Corporation (Billerica, MA, USA) for energy-dispersive X-ray spectroscopy (EDS).

## 3. Results

### 3.1. The Characteristics of Mn Sludge

Table 1 displays the XRF analysis for the Mn sludge, showing a significant amount of Mn. Moreover, Figure 1 illustrates the XRD spectrum for the Mn sludge, which indicates that the Mn exists mainly in the form of a carbonate compound as rhodochrosite (MnCO_3_). Previous studies have also reported the presence of MnCO_3_ in this type of sludge [5]. Therefore, in the XRF analysis presented in Table 1, this compound was specifically considered, while the most stable oxides were considered for the other metals. It is important to note that due to the high intensity of the diffracted X-ray signal from MnCO_3_, the identification of other minor phases was not possible.

### 3.2. Leachability of Mn Sludge

#### 3.2.1. Effect of Temperature

Figure 2 presents the measured chemical compositions of the leaching solutions, obtained using 1.7 M H_2_SO_4_ solution for three L/S ratios at 30 °C and 50 °C. The analysis reveals that Mn, Zn, and K are the main elements being dissolved into the solution, followed by Na, Mg, etc. Clearly, higher L/S ratios result in lower concentrations of Mn, Zn, and K in the solutions, indicating that for the lowest L/S ratio, significant portions of these elements are leached. Moreover, increasing the leaching temperature from 30 °C to 50 °C generally leads to a higher leaching rate of the elements.

To gain a better understanding of the temperature effect, the recovery rates of Mn, Zn, and K were calculated for different L/S ratios, as illustrated in Figure 3. The results demonstrate that as the temperature rises from 30 °C to 50 °C, the recovery of metals into the leachates increases, considering the slightly lower accuracy observed for L/S = 5. Therefore, the temperature was fixed at 50 °C for all the other experiments.

#### 3.2.2. Effect of Solution Volume and Concentration

The leaching behavior of the sludge is influenced by the L/S ratio and acid concentration, as displayed in Figure 4. It is worth mentioning that the L/S ratio during the leaching process was not constant due to gas release via the chemical reactions involved and the formation of bubbles. Hence, the L/S ratio values that are presented are the numbers at initial. It is evident that increasing the L/S ratio for a given acid concentration leads to a higher leaching rate, with Mn and Zn recoveries reaching approximately 90% and 70%, respectively. Figure 4a indicates that different acid concentrations result in the leaching of most of the Mn. Higher acid concentrations lead to increased Mn leaching rates at fixed leaching durations for each L/S ratio. However, for higher L/S ratios, the Mn leaching rate becomes less dependent on the acid concentration. A similar trend is observed for the leaching of Zn from the sludge (Figure 4b), where 66% to 77% of Zn is dissolved using 0.7 M to 1.7 M acid solutions. Figure 4c shows that the dissolution of K during the leaching process is consistently higher than that of Mn and Zn, following the same trend with changes in the L/S ratio. Moreover, the differences in the K leaching rates for different acid concentrations at a given L/S ratio are smaller compared to Mn and Zn leaching rates.

#### 3.2.3. Effect of Leaching Duration

Figure 5 shows the effect of leaching duration on the dissolution of Mn (a), Zn (b), and K (c) from the sludge under fixed temperature and L/S ratio. The experimental data consistently indicate a rapid initial stage of leaching, followed by a much slower stage.

As illustrated in Figure 5, a significant portion of Mn, Zn, and K is leached within the first 10 min. Furthermore, the initial leaching rate is high and not significantly dependent on the acid concentration. However, the leaching extent is higher for the 1.2 M solution compared to the 1.7 M solution. Nevertheless, this difference becomes insignificant after 120 min of leaching. Obtaining a lower leaching rate for Mn and Zn with 1.2 M solution in the sample leached for 120 min than that leached for 90 min is difficult to explain, and the measured concentrations may be outliers, thus further experimental work is needed to clarify this area.

The pH measurements reveal an increase in the solution pH during the leaching process, with greater pH changes observed when lower acid volumes are used. For example, when using a 1.7 M solution, the pH increases from approximately −0.23 to approximately 5 for L/S ratios of 1.25 and 2.5, while it reaches approximately 5 for an L/S ratio of 5. In several experiments using a 1.7 M solution at L/S = 10, the pH increases within a range from 0.16 to 0.44. The final pH for 0.7 and 1.2 M H_2_SO_4_ concentrations is also approximately five. However, since the pH changes were not continuously measured, they cannot be used to study the leaching rate.

### 3.3. The Phase Analysis of Residue

Figure 6 displays the XRD spectra of the leaching residues obtained from experiments conducted at different leaching durations using a 1.2 M H_2_SO_4_ solution and an L/S ratio of 10 in comparison to the Mn sludge. The results show that during the leaching process, MnCO_3_ from the sludge is leached, while lead sulfate (PbSO_4_) simultaneously forms. The XRD analysis data presented in Figure 6 confirm that the presence of other crystalline residue components is not significant compared to these two phases.

The identified phases in the leaching residues obtained using various acid concentrations and L/S ratios are shown in 7. When a 0.7 M acid solution is used, a significant amount of MnCO_3_ remains unbleached, even at higher L/S ratios, which is consistent with the previously presented results regarding the recovery of Mn into the solutions. Comparing the XRD spectra in Figure 7 indicates that the PbSO_4_ phase appears in the sludge leached at L/S = 2.5 when a 1.7 M acid solution is used, while for lower acid concentrations, PbSO_4_ is observed at L/S = 5. Moreover, for this L/S ratio, more intense PbSO_4_ peaks and correspondingly weaker MnCO_3_ peaks are observed when a stronger acid is used. The XRD spectra of the residue samples obtained from leaching conditions with L/S = 10 and a 1.7 M acid solution were quite similar to the sample leached with a 1.2 M solution for L/S = 10, presented in Figure 6, and they all exhibit PbSO_4_ as the main phase.

### 3.4. The Microstructural Analysis of Residue

Figure 8 presents the elemental X-ray mapping of a representative leaching residue, demonstrating a high Mn recovery rate of approximately 89%. The figure illustrates that, during the leaching process, both small particles and relatively large agglomerates can be formed. The agglomerates exhibit a significant amount of Pb and S, which are closely associated, confirming the dominance of the PbSO_4_ phase as the main compound in the residue. Additionally, a distinct layer of PbSO_4_ is observed on the surface of the agglomerates. The distribution of Mn, Fe, Al, Si, and Ca within the sample is not uniform, whereas Zn, K, and Na are observed to be distributed throughout the sample.

The distribution of Mn, Fe, and O as well as the analysis of point 1 in Figure 8 indicates that Fe, Mn, and O are accumulated in some particles. Similarly, the distribution of Al, Si, and O, as observed in point 3, demonstrates a similar pattern for these elements. On the other hand, the analysis of point 2 reveals the coexistence of all major elements, including Pb, Zn, Fe, Mn, S, and O, in certain areas of the sample.

## 4. Discussion

### 4.1. Mn Sludge Composition

Obtaining an overview of the composition and formation of Mn sludge components is important to understand the leaching behavior of the sludge. The minor oxides components of sludge, such as SiO_2_, Al_2_O_3_, CaO, and MgO (Table 1) originated from the SAF charge via the transportation of fine particles (dust) by the furnace gas. However, Mn, Zn, and K components end up in the sludge via a different mechanism with chemical reactions involved. Mn in the sludge was found in the form of MnCO_3_, as shown in Figure 1, whereas Mn ores used in the ferroalloy industry are typically in the form of oxides (e.g., MnO_2_, Mn_2_O_3_) or silicates (e.g., MnSiO_3_). The presence of MnCO_3_ in the submerged arc off-gas furnace can be attributed to the significant amounts of CO and CO_2_ gases generated during the furnace operation. These gases are produced from the charged solid carbon reductant through the carbothermic reduction of MnO by C, the gaseous reduction of higher Mn oxides by CO gas, and the Boudouard reaction, as discussed previously [2]. In addition, carbonate compounds present in the furnace charge, such as CaCO_3_, MgCO_3_, and K_2_CO_3_ are thermally decomposed at elevated temperatures, resulting in the generation of CO_2_ gas within the furnace burden. Given the significant amount of CO in the SAF gas outlet, it is expected that the Mn-containing compound leaving the furnace is mainly in the form of MnO at temperatures ranging from 200 to 400 ℃. The decrease in gas temperature causes an increase in the CO_2_ content, leading to the conversion of MnO to MnCO_3_. This change in gas composition results from the conversion of CO to CO_2._ Consequently, MnCO_3_ is formed through the following reaction:(1)$$\mathrm{MnO}+{\mathrm{CO}}_{2}={\mathrm{MnCO}}_{3}$$

By calculating the changes in Gibbs energy with temperature for various CO_2_ partial pressures for chemical reaction (1), it was found that the formation of MnCO_3_ occurs at temperatures below 340 °C, 335 °C, 325 °C, 320 °C, and 300 °C for CO_2_ partial pressures of 1 atm, 0.8 atm, 0.6 atm, 0.4 atm, and 0.2 atm, respectively. Clearly, the reaction temperature is not significantly affected by the CO_2_ partial pressure, and the reaction proceeds at temperatures below 300 °C. Similar reactions may take place for some minor components present in the dust, such as K and Zn, leading to the formation of their carbonates. The presence of these phases was investigated previously [5]. The studied sludge contained a high amount of Zn, which could be present in the forms of ZnO and ZnCO_3_. However, these compounds were not detected in the XRD analysis, possibly because ZnO exists in an amorphous form, resulting from its formation from the gas phase. The amount of K in the sludge is much lower than Mn and Zn, while higher leaching rates for K are observed in comparison with Mn and Zn (Figure 5). This may indicate that it has a higher reactivity with sulfuric acid and a large portion of it is dissolved rapidly. It is noted that this may confirm good contact between the solid fine particles with the solution during the leaching process.

### 4.2. The Kinetics of Leaching

Mn and Zn are the most valuable metals in the sludge due to their abundance and leachability. The leaching process of these elements from the sludge takes place in two stages: an initial fast stage followed by a slower secondary stage, as shown in Figure 5. To determine the reaction rate constant, the conversion fractions of MnCO_3_ to Mn^2+^ (α) and ZnO to Zn^2+^ were calculated based on the leachability data for 1.2 and 1.7 M solutions at different durations. The α-values were tested for different chemical reaction orders, and it was found that the first-order reaction does not fit the leaching process for Mn and Zn. However, the second and third-order reactions, described by Equations (2) and (3), respectively, provide a better fit to the data, as demonstrated in Figure 9, with the exception of one outlier measurement for the 1.7 M solution. It is important to note that the vertical axis values in Figure 9 represent the α functions given by the following equations:(2)$$F_{2}\left(\alpha \right)=\frac{1}{1-\alpha }-1=k_{2}t,$$
(3)$$F_{3}\left(\alpha \right)=0.5\left(\frac{1}{{\left(1-\alpha \right)}^{2}}-1\right)=k_{3}t$$
where *k*_2_ and *k*_3_ are the reaction rate constants, and *t* is the leaching time.

Based on the above equations, the slope of the fitted lines in Figure 9 represents the reaction rate constants *k*_2_ and *k*_3_. These values were calculated and presented in Table 2. The obtained rate constant values show that the leaching rate of Mn is 3 to 6 times higher for the 1.2 M solution compared to the 1.7 M solution. Similarly, the leaching rate of Zn is 4 to 15 times higher for the 1.2 M solution compared to the 1.7 M solution. This observation may explain the reason for the higher leaching rate of Mn in the 1.2 M solution compared to the 1.7 M solution for an L/S ratio of 10, as shown in Figure 4 and Figure 5. Furthermore, comparing the rate constant values for Mn and Zn leaching indicates that the leaching of Mn is 2 to 5 times faster than the leaching of Zn. However, the difference in leaching rates between Mn and Zn is greater for the 1.7 M solution compared to the 1.2 M solution.

### 4.3. Thermochemistry of Reactions

The results showed a significant mass transport of Mn and Zn from the sludge into the solution during leaching. The leaching chemistry of these elements is discussed as follows.

#### 4.3.1. Mn Leaching

It was observed that the majority of Mn in the sludge exists in the form of MnCO_3_. During the leaching process, Mn is leached, resulting in an increase in pH and the consumption of acid. On the other hand, significant gas evolution occurs during leaching, suggesting the following reaction:(4)$$MnCO_{3}\left(s\right)+H_{2}SO_{4}\left(aq\right)=MnSO_{4}\left(aq\right)+H_{2}O\left(l\right)+CO_{2}\left(g\right).$$

Clearly, the generation of CO_2_ gas within the system is the main mechanism for bubble formation. The formation of bubbles combined with the applied magnetics stirring may explain the observed rapid leaching rates of Mn, as shown in Figure 5 The Eh–pH diagram of the Mn–S–H_2_O system, depicted in Figure 10 using FactSage thermodynamic software, illustrates that Mn^2+^ and MnSO_4_(aq) are formed during leaching, considering the above reaction. The final pH of the solution was measured within the range of 0.16 to 5, indicating that the solution is within the Mn^2+^ and MnSO_4_(aq) region.

The highest obtained Mn leaching rate of about 90% (Figure 4 and Figure 5), the shape of the leaching curves in Figure 5, and observing no MnCO_3_ phase in the samples leached by 1.2 M and 1.7 M solutions at L/S = 10 collectively indicate that the majority of MnCO_3_ has been leached. It is possible that the remaining Mn has not had proper contact with the acid due to the formation of PbSO_4_ on sludge agglomerates (Figure 8) that causes a further reduction in the Mn leaching rate, a kind of passivation. The other explanation is that a small portion of Mn (less than 10 wt%) exists in other amorphous minor phases (not MnCO_3_) that were not observed in XRD spectra and such phases may not be leachable.

#### 4.3.2. Zn Leaching

As mentioned earlier, Zn in the sludge exists in the form of ZnO and ZnCO_3_. In the presence of an H_2_SO_4_ solution, these compounds dissolve through the following reactions:(5)$$ZnO\left(s\right)+H_{2}SO_{4}\left(aq\right)=ZnSO_{4}\left(aq\right)+H_{2}O.$$
(6)$$ZnCO_{3}\left(s\right)+H_{2}SO_{4}\left(aq\right)=ZnSO_{4}\left(aq\right)+H_{2}O\left(l\right)+CO_{2}\left(g\right).$$

The Eh–pH diagram for the Zn–S–H_2_O system in Figure 11 illustrates that the aqueous ZnSO_4_ remains in a stable phase within the range of final pH values measured between 0.16 and 5. Consequently, the chemical reaction (2) takes place, leading to the dissolution of Zn in the sludge. Like chemical reaction (2), the formation of CO_2_ gas through chemical reaction (2) may contribute to the formation of gas bubbles during leaching.

Chemical reactions (5) and (26) exhibit significant progress when using high concentrations of acids and higher L/S ratios. For example, when employing 1.2 M and 1.7 M acid solutions with an L/S ratio of 10, the acid content is sufficient to leach Mn and Zn significantly. However, when using a 1.2 M acid solution with L/S ratios of 1.25 and 2.5, it is theoretically possible to leach up to 30% and 60% of Mn, respectively. Nevertheless, due to the simultaneous leaching of Zn and K, the leaching of Mn is lower than the theoretical values. On the other hand, as the system approaches equilibrium, the leaching does not proceed further. This explains the existence of unbleached sludge at lower L/S ratios, as depicted in Figure 7. Based on the obtained results, the best leaching conditions can be achieved by using acid solutions with concentrations of 1.2 M and higher in combination with high L/S ratios.

Figure 5 indicates that the initial leaching rates of Mn and Zn with 1.2 M and 1.7 M solutions are similar for the L/S = 10. However, the extent of leaching in the initial stage and further leaching kinetics in the second slow stage are higher for the 1.2 M solution than the 1.7 M solution. These differences may indicate that the concentration of sulfuric acid reactants above 1.2 M solution is not important for the process kinetics. Hence, we may suggest that the leaching process is more dependent on the mass transport of the chemical reaction products away from the reaction surface. Obviously, the mass transport of MnSO_4_(aq) and ZnSO_4_(aq) is faster in the more dilute solution of 1.2 M solution than that for the 1.7 M solution, and this may explain the above observations.

#### 4.3.3. Lead Sulfate Formation

In all the leaching experiments, the formation of solid PbSO_4_ and the insignificant dissolution of Pb into the solution were observed. This can be attributed to the stability of solid PbSO_4_ in the system, as predicted by the Eh–pH diagram for the Pb–S–H_2_O system in Figure 12. The final pH values of all the experiments ranged from 0.16 to approximately 5, within which solid PbSO_4_ remains in a stable phase.

In the presence of sulfuric acid with the sludge, the formation of PbSO_4_ can be explained by the following reaction:(7)$$PbO\left(s\right)+H_{2}SO_{4}\left(aq\right)=PbSO_{4}\left(s\right)+H_{2}O.$$

The complete conversion of PbO to PbSO_4_ during leaching is evident from the intense peaks of PbSO_4_ observed in the XRD spectra of several residue samples. Clearly, the other components of unbleached sludge are found in significantly lower amounts in samples with high recoveries of Mn and Zn. The extensive progress of reaction (2) can be attributed to the particle size of PbO in the sludge, which is expected to be very fine. These particles likely condense from the gas phase as Pb gas exits SAF and subsequently oxidizes to PbO (which has a high melting point of 888 °C) before condensation.

## 5. Conclusions

The leaching behavior of a manganese sludge using sulfuric acid solutions was investigated, and the main conclusions are as follows:The leaching rate of Mn sludge is fast, resulting in high leaching rates of 90% for both Mn and Zn.The leaching rates of Mn, Zn, and K increase with increasing temperature, liquid to solid ratio, and duration. Mn exhibits a higher maximum leaching rate compared to Zn.Comparing acid solutions, the leaching rates of Mn and Zn are slightly higher using a 1.2 M H_2_SO_4_ solution compared to a 1.7 M H_2_SO_4_ solution, while the leaching rates are much lower with a 0.7 M acid solution.Kinetic studies of leaching with 1.2 M and 1.7 M H_2_SO_4_ acid solutions reveal that the reaction rate constant for Mn dissolution exceeds 1.2 × 10^−3^ s^−1^, while for Zn dissolution, it is higher than 5 × 10^−4^ s^−1^.During the leaching process, aqueous sulfate solutions of MnSO_4_ and ZnSO_4_ are formed from the dissolution of solid MnCO_3_ and ZnO (or ZnCO_3_) present in the sludge. This process is accompanied by the generation of CO_2_ gas and bubble formation.The Pb content in the sludge is fully converted to solid PbSO_4_ during leaching, and samples with high Mn and Zn leaching rates yield a residue of lead sulfate.

## Figures and Tables

**Figure 1 materials-16-04591-f001:**
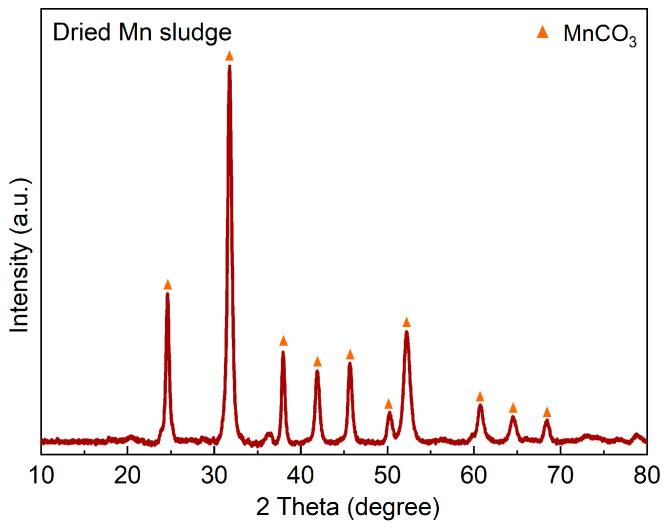
XRD spectrum of the dried Mn sludge with the identified phase.

**Figure 2 materials-16-04591-f002:**
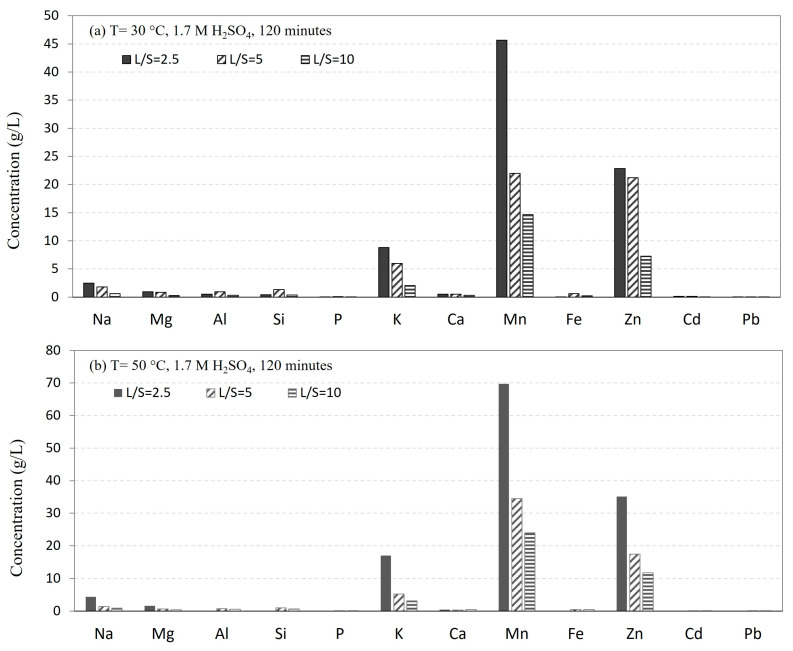
The concentrations of elements in solutions after leaching at different L/S ratios and temperatures using 1.7 M H_2_SO_4_ solution at (**a**) 30 °C and (**b**) 50 °C.

**Figure 3 materials-16-04591-f003:**
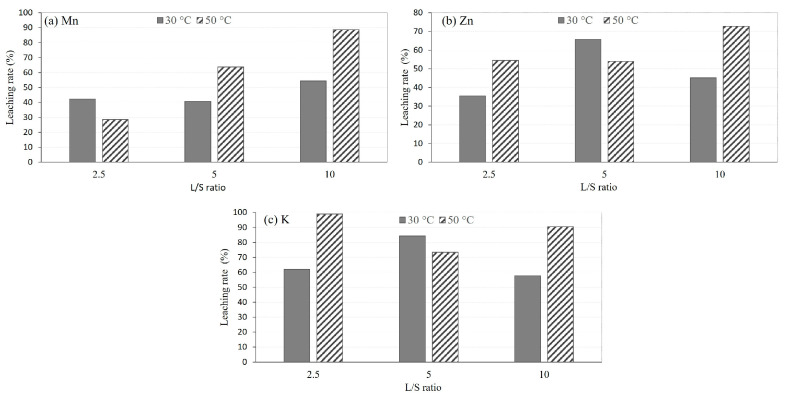
The leaching rates for different temperatures and L/S ratios of (**a**) Mn, (**b**) Zn, and (**c**) K based on XRF measurements.

**Figure 4 materials-16-04591-f004:**
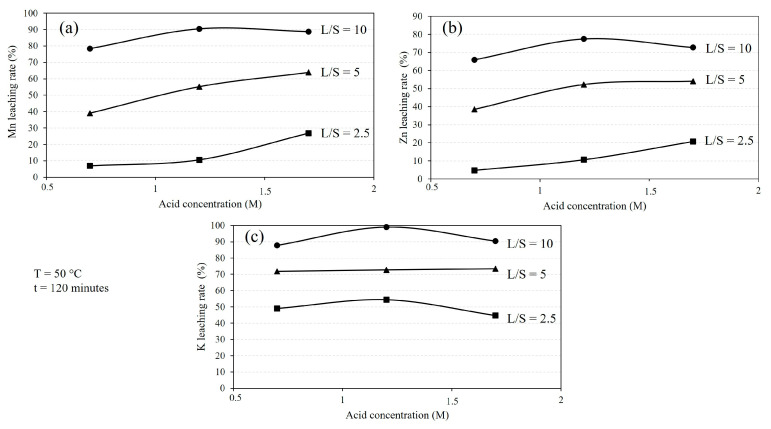
The effect of L/S ratio and acid concentration on the leaching rates of (**a**) Mn, (**b**) Zn, and (**c**) K at 50 °C and duration of 120 min, based on XRF measurements.

**Figure 5 materials-16-04591-f005:**
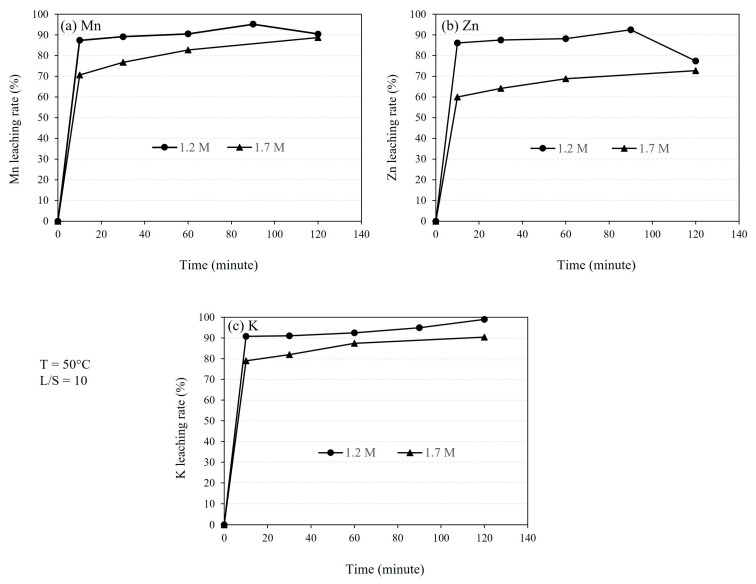
The leaching rates of (**a**) Mn, (**b**) Zn, and (**c**) K using different acid concentrations, namely 1.2 M and 1.7 M H_2_SO_4_ solutions at 50 °C for L/S ratio of 10.

**Figure 6 materials-16-04591-f006:**
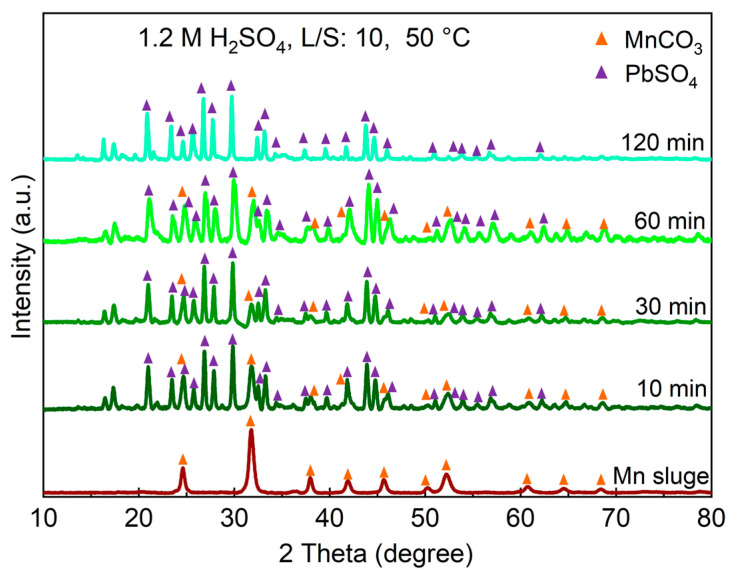
The XRD spectra of leaching residue samples at different leaching durations using 1.2 M H_2_SO_4_ solution and an L/S of 10 at 50 °C.

**Figure 7 materials-16-04591-f007:**
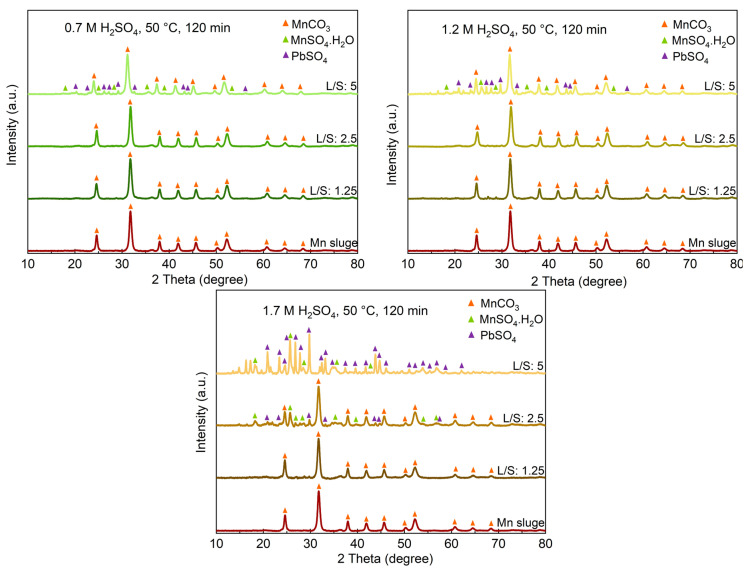
The XRD spectra of leaching residues obtained in leaching with different acid concentrations and L/S ratios.

**Figure 8 materials-16-04591-f008:**
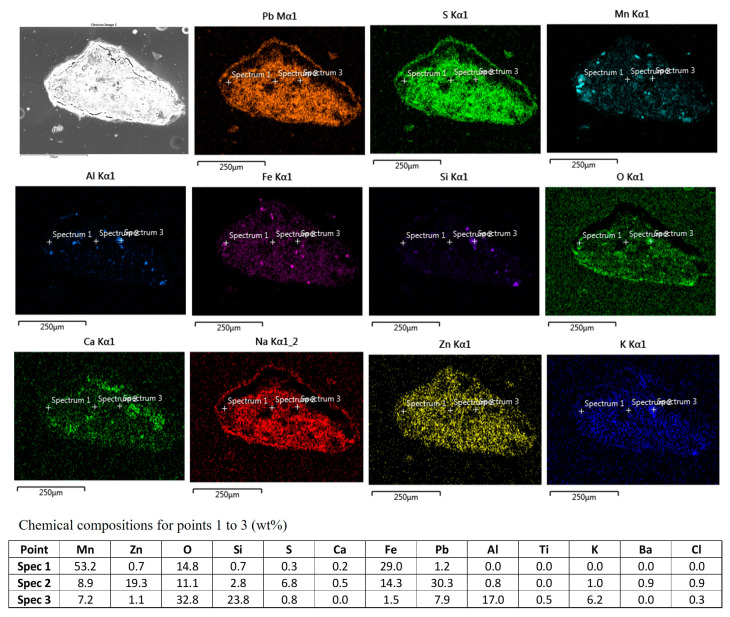
The X-ray mapping of elements in an agglomerate of the leaching residue, and EDS point analysis for selected particles.

**Figure 9 materials-16-04591-f009:**
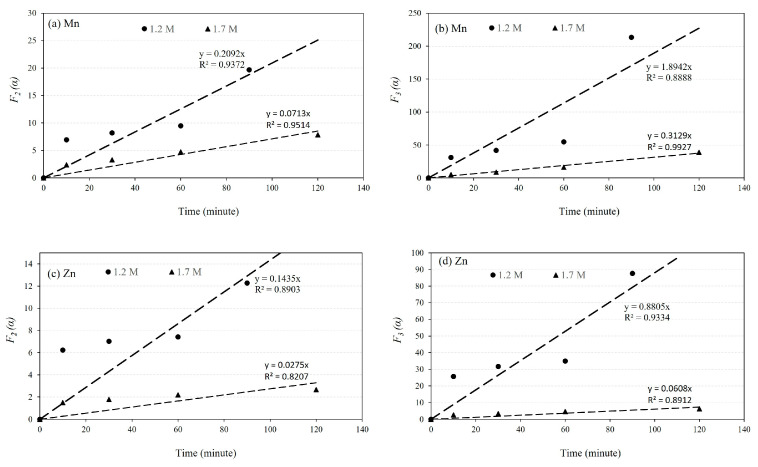
The relationship between *F*_2_ (*α*) and *F*_3_ (*α*) with leaching time for two different acid concentrations of 1.2 and 1.7 M. Mn: (**a**,**b**), Zn: (**c**,**d**).

**Figure 10 materials-16-04591-f010:**
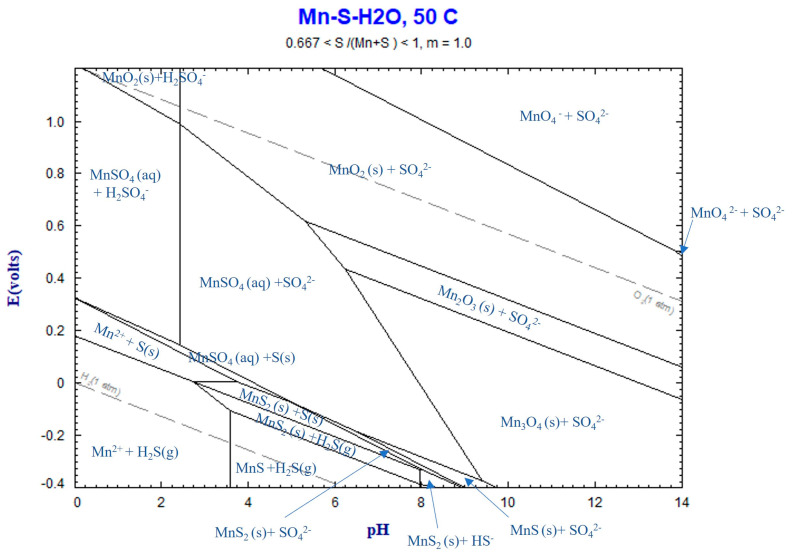
Eh––pH diagram for Mn–S–H_2_O system at 50 °C, calculated by FactSage thermodynamic software version 8.2.

**Figure 11 materials-16-04591-f011:**
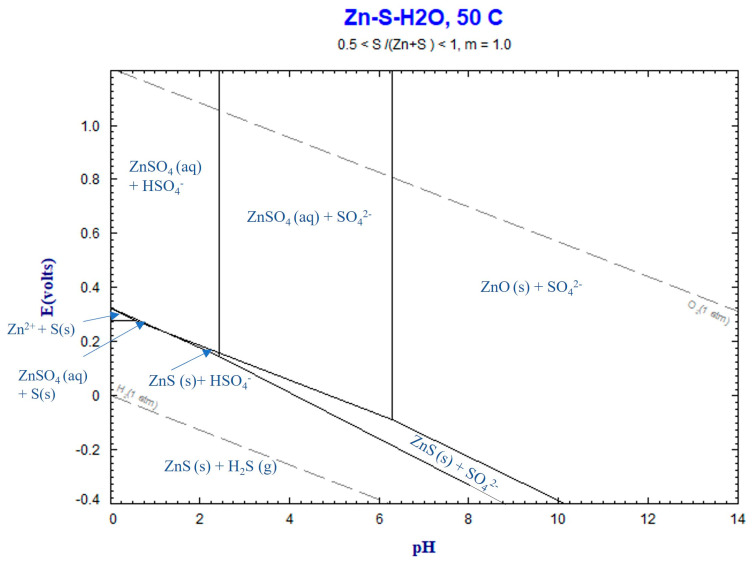
Eh–pH diagram for Zn–S–H_2_O system at 50 °C, calculated by FactSage thermodynamic software version 8.2.

**Figure 12 materials-16-04591-f012:**
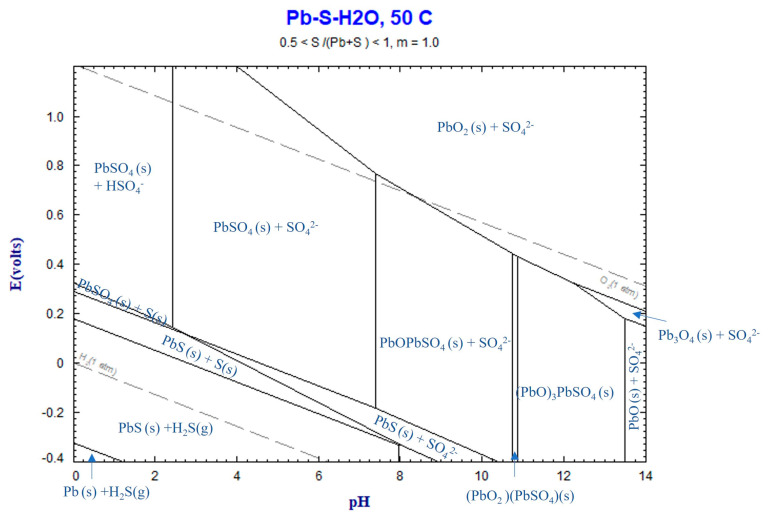
Eh–pH diagram for Pb–S–H_2_O system at 50 °C, calculated by FactSage thermodynamic software version 8.2.

**Table 1 materials-16-04591-t001:** Chemical composition of Mn sludge measured by XRF (wt%).

MnCO_3_	Fe_2_O_3_	MgO	SiO_2_	Al_2_O_3_	CaO	TiO_2_	BaO	ZnO	PbO	Na_2_O	K_2_O	P_2_O_5_	CdO	SO_3_
63.65	1.92	0.61	1.75	1.14	0.96	0.03	0.09	20.08	1.66	2.88	4.28	0.09	0.09	0.79

**Table 2 materials-16-04591-t002:** Rate constant values for Mn and Zn leaching from the sludge for different acid concentrations.

Solution Concentration	Rate Constant *k*_2_ (s^−1^)		Rate Constant *k*_3_ (s^−1^)	
	Mn	Zn	Mn	Zn
1.2 M	3.5 × 10^−3^	2.4 × 10^−3^	3.16 × 10^−2^	1.47 × 10^−2^
1.7 M	1.2 × 10^−3^	5 × 10^−4^	5.2 × 10^−3^	1.0 × 10^−3^

## Data Availability

Data is available in a publicly accessible repository that does not issue DOIs. Publicly available datasets were analyzed in this study. This data can be found at https://www.ntnu.edu/metpro (accessed on 20 May 2023).

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
