# Peer review of "The Effect of Process Conditions on Sulfuric Acid Leaching of Manganese Sludge"

_materials, 2023, doi:10.3390/ma16134591_

Round 1

Reviewer 1 Report

The input material consists of manganese ore dusts, which are produced during the production of manganese alloys by carbothermal reduction in an arc furnace (SAF).

The leaching process was carried out under specified conditions. Sulfuric acid was used as leaching agent. Temperature, acid concentration, volume of leaching agent, leaching time, L/S ratio were varied. After the leaching process, the sample was filtered and the pH was measured.

 From the experiments conducted, we can determine that the leaching rate of Mn, Zn and K increases with increasing temperature, liquid to solid ratio and duration. Mn shows a higher maximum leaching rate than Zn.

- During the leaching process, aqueous solutions of MnSO4 and ZnSO4 sulphates are formed by dissolving the solid MnCO3 and ZnO (or ZnCO3) present in the sludge.

For leaching conditions, I recommend experimentally verifying the use of hydroxide as a leaching agent, why just acid, try hydroxide which transitions zinc well into leach.

Why the specified leaching parameters were chosen - there is no explanation.

Improve the quality of figure 3

Author Response

Dear reviewer,

I would appreciate your comments and recommendations to improve the manuscript, they were found constructive. The responses to your comments are provided as follows:

Reviewer comment 1: For leaching conditions, I recommend experimentally verifying the use of hydroxide as a leaching agent, why just acid, try hydroxide which transitions zinc well into leach.

Why the specified leaching parameters were chosen - there is no explanation.

Response: In literature different leaching agents can be used to leach Mn sludge, and a few literature works using other acid types were mentioned in section 1, using alkali solutions has not been studied. To clarify, the following sentence was added into the introduction part (section 1), 3rd line in the last paragraph:

“In hydrometallurgical processes to produce Mn and Zn metals sulfuric acid has been used as the leaching agent. Since the Mn sludge has significant amount of these two metals, in the present study sulfuric acid leaching was considered as the leaching agent to use.”

Reviewer comment 2: Improve the quality of figure 3

Response: The quality of Figure 3 was improved as recommended.

Best Regards,

Jafar Safarian (the corresponding author)

Reviewer 2 Report

This is a meaningful work for the valorization of Mn sludge. In general, the work is well designed, conducted and presented. I recommend to publish this work after a minor revision. Some key comments are as follows:

1. In 2.1 Materials and preparation section, the grade of sulfuric acid should be added.

2. For the investigation of liquid to solid ratio, it seems that the volume of sulfuric acid and the amount of sludge powder were fixed while the total volumes of leaching were regulated. In other words, a higher liquid to solid ratio resulted in a lower concentration of the sulfuric acid. This is a little different from our common sense. I recommend the authors to adjust corresponding description.

3. The sentences in lines 118, 132-133 are meaningless, which should be deleted.

4. According to the results presented in Table 1 and Figure 1, the addition of a word of “mainly” is needed in line 122.

5. Error bar of the data should be added in all Figures 2-5.

6. In abstract section, the authors claimed that the leaching rates of Mn, Zn and K increased with liquid to solid ratio. Unfortunately, the result of Figure 3 dose not seem to support this.

7. The leaching rates of Mn and Zn decreased after 90 min in Figure 5, a reasonable explanation is needed.

8. The format of the reference [15] is not standardized.

Author Response

Dear reviewer,

I would appreciate your comments and recommendations to improve the manuscript, they were found constructive. The responses to your comments are provided as follows:

Comment 1. In 2.1 Materials and preparation section, the grade of sulfuric acid should be added.

Response: The acid grade was added in the end of 1st paragraph in section 2.1:

“Concentrated sulfuric acid (98.3%H2SO4) was diluted.”

Comment 2. For the investigation of liquid to solid ratio, it seems that the volume of sulfuric acid and the amount of sludge powder were fixed while the total volumes of leaching were regulated. In other words, a higher liquid to solid ratio resulted in a lower concentration of the sulfuric acid. This is a little different from our common sense. I recommend the authors to adjust corresponding description.

Response: Thanks for your comment, in the experimental work, regarding the applied mass of solid (10 grams), the volume of acid was used 12.5, 25, 50, and 100 mL. It is very common to have L/S ratio in form of L-solution /g-solid. This is clearly described in 2nd to 4th sentences in 1st paragraph of section 2.2.

Comment 3. The sentences in lines 118, 132-133 are meaningless, which should be deleted.

Response: “They were removed as recommended”

Comment 4. According to the results presented in Table 1 and Figure 1, the addition of a word of “mainly” is needed in line 122.

Response: Thanks for the suggestion, the word “mainly” was added in this sentence as recommended.

Comment 5. Error bar of the data should be added in all Figures 2-5.

Response: Thanks for your comment! We agree that from scientific point of view it is better to add the error bars. However, only one sample was analyzed by other labs using XRF and ICP-MS techniques (no parallels). This was due to analyzing many samples in this work and not possibility of using many parallels and doing statistical work.

Comment 6. In abstract section, the authors claimed that the leaching rates of Mn, Zn and K increased with liquid to solid ratio. Unfortunately, the result of Figure 3 dose not seem to support this.

Response: Thanks for your comment! To prevent confusion, the word “overall” was added in the related sentence in abstract. The overall means the total amount of Mn+Zn+K into the solution.

“The results revealed that Mn, Zn, and K are the main leachable components, and their overall leaching rates increase with increasing temperature, liquid to solid ratio, and time”

Comment 7. The leaching rates of Mn and Zn decreased after 90 min in Figure 5, a reasonable explanation is needed.

Response: We agree, the following sentence was added at the end of second paragraph of section 3.2.3:

“Obtaining lower leaching rate for Mn and Zn with 1.2 M solution in the sample leached for 120 minutes than that leached for 90 minutes is difficult to explain and the measured concentrations may be outlier, and further experimental work is needed to clarify. “

Comment 8. The format of the reference [15] is not standardized.

Response: Thanks, the reference format was updated and corrected.

Best Regards,

Jafar Safarian (the corresponding author)

Reviewer 3 Report

Article is well written. Some minor corrections are suggested as follows:

L 80 -90 %: in which reference was it stated – add reference.

3.2.2 Argue measurements:

Dissolution of K during the leaching process is consistently higher? Why destructure of K, why such trends?

 In Figure 9: which represent what: e.g. 9a and 9b - what is the difference? Explain in Figure capture.

 Figure 11 and Figure12: It is not clear if they were done based on Your measurements and calculations or who's? Explain. If Your calculations are presented, how does it differ/agree with other studies and why so? Discuss.

Explain, why is the leaching rate of Mn and Zn is slightly higher using a 1.2 M than 1.7 M acid?

 The number of references is low: add some related researches and discuss Your and their results, what is added value of present study?

References are not cited according to guidelines: e.g. 11,14,15: are these articles or not? Please cite more precise.

English is good.

Author Response

Dear reviewer,

I would appreciate your comments and recommendations to improve the manuscript, they were found constructive. The responses to your comments are provided as follows:

Comment 1: Article is well written. Some minor corrections are suggested as follows:

L 80 -90 %: in which reference was it stated – add reference.

Response: I hope I understod well your comment, the following was added in the end of 1st paragraph in section 2.1:

“Concentrated sulfuric acid (98.3%H2SO4) was diluted.”

Comment 2: 3.2.2 Argue measurements:

Dissolution of K during the leaching process is consistently higher? Why destructure of K, why such trends?

Response: We measured the solutions concentrations and obviously K (K2O or K2CO3,...) is leached faster and more by H2SO4 compared to Mn and Zn. The presented data are with reference to the initial content of K, which is much lower than Mn and Zn, and the high leaching rates for K indicates that it has high reactivity with sulforic acid. The following sentence was added in discussion section at the end of last paragraph of section 4.1:

“The amount of K in the sludge is much lower than Mn and Zn, while higher leaching rates for K is observed in comparison with Mn and Zn (Fig. 5). This may indicate that it has a higher reactivity with sulfuric acid and a large portion of it is getting dissolved rapidly. It is noted that this may confirm good contact between the solid fine particles with the solution during the leaching process.”

 Comment 3: In Figure 9: which represent what: e.g. 9a and 9b - what is the difference? Explain in Figure capture.

Response: The figure caption was updated:

“The relationship between F2 (α) and F3 (α) with leaching time for two different acid concentrations of 1.2 and 1.7 M, Mn: a and b, Zn: c and d. “

Comment 4: Figure 11 and Figure12: It is not clear if they were done based on Your measurements and calculations or who's? Explain. If Your calculations are presented, how does it differ/agree with other studies and why so? Discuss.

Response: The calculations by FactSage were carried out considering the solution concentrations close to the present study and as displayed in the figures for 1 molal solutions and sulfur concentration ranges valid for the applied conditions in this work. Unfortunately no literature work on the same material was found to compare with.  

Comment 5: Explain, why is the leaching rate of Mn and Zn is slightly higher using a 1.2 M than 1.7 M acid?

Response: Thanks for this comment. The following short paragraph text was added at the of section 4.3.2 to explain this observation:

“Figure 5 indicates that the initial leaching rates of Mn and Zn with 1.2 M and 1.7 M solutions are similar for the L/S = 10. However, the extent of leaching in the initial stage, and further leaching kinetics in the second slow stage are higher for 1.2 M solution than 1.7 M solution. These differences may indicate that the concentration of sulfuric acid reactant above 1.2 M solution is not important for the process kinetic. Hence, we may suggest that the leaching process is more depending on the mass transport of the chemical reaction products away from the reaction surface. Obviously, the mass transport of MnSO4(aq) and ZnSO4(aq) is faster in the more dilute solution of 1.2 M solution than that for 1.7 M solution, and this may explain the above observations. “     

 Comment 6: The number of references is low: add some related researches and discuss Your and their results, what is added value of present study?

Response: Unfortunately we could not find more references in the area of Mn sludge/dust processing. This research area is getting now more attention regarding circular economy and sustainable development.

Comment 7: References are not cited according to guidelines: e.g. 11,14,15: are these articles or not? Please cite more precise.

Response: The references were cited more precisely, reference 11 has been cited several times with the same format.

Best Regards,

Jafar Safarian (the corresponding author)

Round 2

Reviewer 1 Report

For Figures 3 and 4, I would recommend stating the method of measurement, although this is given in Chapter 2.

I would recommend deleting line 243. "The obtained results are discussed in more detail."

Author Response

Dear reviewer,

Thank you very much for the comments, they were applied. 

1- The measurement method was added into the captions for figures 3 and 4: "based on XRF measurements"

2- The line 243 was removed.

Best Regards,

Jafar Safarian (the corresponding author)